# Experimental Data of a Floating Cylinder in a Wave Tank: Comparison Solid and Water Ballast

**Roman Gabl [1,2,*]** [ID], **Thomas Davey [1]** [ID], **Edd Nixon [1]**, **Jeffrey Steynor [1]** and **David M. Ingram [1]** [ID]

[1]  School of Engineering, Institute for Energy Systems, FloWave Ocean Energy Research Facility,
    The University of Edinburgh, Max Born Crescent, Edinburgh EH9 3BF, UK;
    tom.davey@flowave.ed.ac.uk (T.D.); Edward.Nixon@sintef.no (E.N.); Jeff.Steynor@ed.ac.uk (J.S.);
    david.ingram@ed.ac.uk (D.M.I.)
[2]  Unit of Hydraulic Engineering, University of Innsbruck, Technikerstraße 13, 6020 Innsbruck, Austria
[*]  Correspondence: roman.gabl@ed.ac.uk

**Abstract:** The experimental set-up allows for the comparison of two different ballast options of a floating cylinder in a wave tank. Four different internal water drafts are tested as well as an equivalent solid ballast option. The model is excited by regular waves, which are characterised with five wave gauges in front of the floating cylinder and two behind. Additionally, the time series of the six-degree freedom response of the floating structure is made available. Regular waves with an initial amplitude of 0.05 m and frequencies over the range 0.3 to 1.1 Hz are investigated. This results in a wide range of different responses of the floating structure as well as very big rotations of up to 20 degrees. This dataset allows for identification of the influence caused by the sloshing of the interior water volume and can be used to validate numerical models of fluid–structure–fluid interaction.

**Keywords:** floating cylinder; water-filled; motion capturing; wave tank; wave gauges; fluid-structure-interaction

---

## 1. Introduction

Sloshing refers to the movement of a fluid in a container or another object with a free surface. Investigations are conducted to predict the free surface run-up heights and pressure acting at the walls, as well as the influence on the movement of floating structures. A wide range of different validation experiments for sloshing can be found in the literature. For example, Caron et al. [1] investigated a cylinder filled with different water levels on a shaking table and Cruchaga et al. [2] used a rectangular tank with a comparable set-up. More complex motions are possible with a six-degree of freedom motion simulation platform, also known as a hexapod [3–5]. In all these cases, the movement of the wall initialises the sloshing of the internal water.

The presented experimental set-up uses a floating structure to investigate interaction between the the inner and outer water body (fluid–structure–fluid interaction). Similar studies for very large floating structures are available in the literature [6–9], but are typically limited to small motions. Ships containing liquefied natural gas are another comparable application. However, in many cases, these experimental investigations incorporate very complex geometry which is case specific and/or superimposed with complex research questions, e.g., the interaction between two ships [10–12].

The two main outcomes achieved with this experiment are (a) simple geometry to reduce the possible influencing parameters, which also reduces the complexity for numerical modelling, and

(b) large motions that create a significant sloshing influence on the behaviour of the floating container. Therefore, a rotationally symmetrical cylinder was chosen, which gave pitch/roll motions of up to 20°.

The goal of the experimental programme is to identify the influence on the response oscillation caused by the sloshing of the internal body of water. Therefore, the floating cylinder is investigated and the water-filled set-up is compared with a solid case of the same mass distribution. The analysis of the study is presented in Gabl et al. [13] and the full data is available in the DataShare repository of the University of Edinburgh [14]. The study is designed as a validation experiment to asses the capability of a numerical model for fluid–structure–fluid interaction, which will help to design comparable applications (storing facilities, platforms or wave energy converter). Based on the specific scaling laws, the results can also be up scaled from the tank scale [15,16].

## 2. Experimental Set-Up

### 2.1. Investigated Cases

A simplified geometry was chosen to restrict the number of variables in the modelled problem. A rotationally symmetrical design had the big advantage that the wave angle has no significant influence on the model behaviour. Nor did the initial condition of the floating structure when the first wave reaches the model. Figure 1 presents an overview of the experiment with the two investigated ballast options. A hollow cylinder was chosen with an outer diameter of 0.5 m and a total height of 0.5 m. The transparent plastic structure had a wall thickness of 7 mm for the ground plate and 5 mm for the wall. All edges were sharp, and the mooring is described and investigated in Gabl et al. [13]. The cylinder incorporates copper tape wave gauges on the wall and floating balls measured by the motion capturing system for additional measurement of the inside free surface [17].

The investigation was conducted at the FloWave Ocean Energy Research Facility at The University of Edinburgh, which is a unique round wave tank with a diameter of 25 m and water depth 2 m for the upper test volume. The tank has the capability to reproduce complex wave condition from any direction combined with current [18–21].

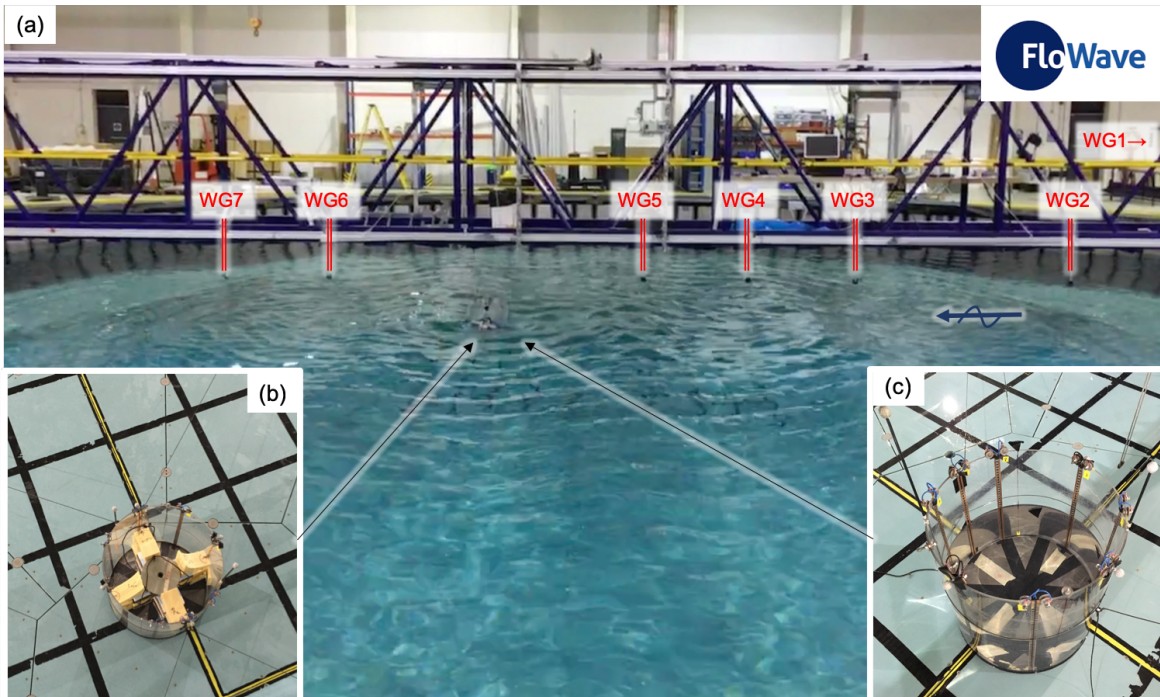

**Figure 1.** Experimental set-up in the tank including (**a**) the wave gauges (WG), (**b**) solid ballast on the raised tank floor and (**c**) variation with an inner water level.

The movement of the body and the free surface markers (six and three degrees of freedom respectively) was captured using a video motion capture system (Section 2.2), with twin wire resistance wave gauges measuring the incident wave conditions (Section 2.3). A trigger signal from the wave-making system was used for data synchronisation.

Figure 1b illustrates the solid ballast configuration on the raised tank floor. This was compared with the fluid case, for which the cylinder was filled with water (Figure 1c) to give an equivalent draft. This was conducted based on the motion capturing system and allows an accuracy of smaller than 1 mm.

Three different solids (two crosses and a cylinder) were used to give a mass distribution in all three axes comparable to the water-filled case. This simplification has only a very small influence on the results as shown by the comparison presented in Gabl et al. [13]. Table 1 lists ballast weights as well as the four combinations as tested. These ballast configurations give a wide range of drafts $d$ (distance between the outside bottom of the cylinder and the still water surface), as determined by the inner water height $h$. The capital letter identifies the individual test case. Table 1 also provides a key between the different drafts and the solid (B, C, D, E) and water (F, G, H, J) ballast options. Letter A was used for the preliminary test to test the position of the wave gauges and wave conditions.Those results are not part of the published dataset. Furthermore, this letter was skipped due to possible confusion. The naming was done according the chronological order of the tests, and it was also used for the provided measurement data. The specific files are described in Section 3.

**Table 1.** Weights of the different components and combination of ballast (Ba1-3); draft $d$ is the distance between still water surface and the bottom of the cylinder, and the variable $h$ indicates the corresponding inner water level depth (Figure 2).

|  | Weight (kg) | d (cm) | h (cm) | Solid | Water |
|---|---|---|---|---|---|
| Cylinder | 6.25 | 3.19 | | | |
| Lead | 5.46 | 2.78 | | | |
| Top | 0.82 | 0.42 | | | |
| dry/empty = CyEmp | 12.52 | 6.39 | 0.00 | | |
| Ballast | | | | | |
| Ba1 | 8.95 | 4.57 | 4.76 | | |
| Ba2 | 17.8 | 9.08 | 9.46 | | |
| Ba3 | 23.18 | 11.83 | 12.32 | | |
| **Combinations** | | | | | |
| CyEmp+Ba1+Ba2+Ba3 = $d \cdot 1.75$ | 62.45 | 31.87 | 26.53 | B | J |
| CyEmp+Ba2+Ba3 = $d \cdot 1.50$ | 53.50 | 27.30 | 21.78 | C | H |
| CyEmp+Ba1+Ba3 = $d \cdot 1.25$ | 44.65 | 22.79 | 17.07 | D | G |
| CyEmp+Ba3 = $d \cdot 1.00$ | 35.70 | 18.22 | 12.32 | E | F |

## 2.2. Motion Capturing System

The oscillation of the floating body was observed with the motion capturing camera system (Qualisys), which is installed at the FloWave test facility. After the initial definition of the global coordinate system at the beginning of the test series, a refinement calibration of the system is conducted before each of the eight test cases. The accuracy is in the order of 1 mm, and the measurement frequency was set to 128 Hz, which was identical to the wave gauges (Section 2.3). The positive x-direction was orientated in the wave direction and the positive z-axis is vertical upwards. The origin of the reported local coordinate system was always at the height of the still water surface as well as the connection points of the symmetric mooring lines. Figure 2 shows this for the water-filled case on the raised tank floor. A rotation around the z-axis (yaw) does not influence the definition of the other two rotation angles. The postprocessing allows such an uncoupling, which is especially useful for rotational symmetrical geometry. The provided six-degree-of-freedom motions were measured in relation to this local body coordinate system. Further information is provided by Gabl et al. [13].

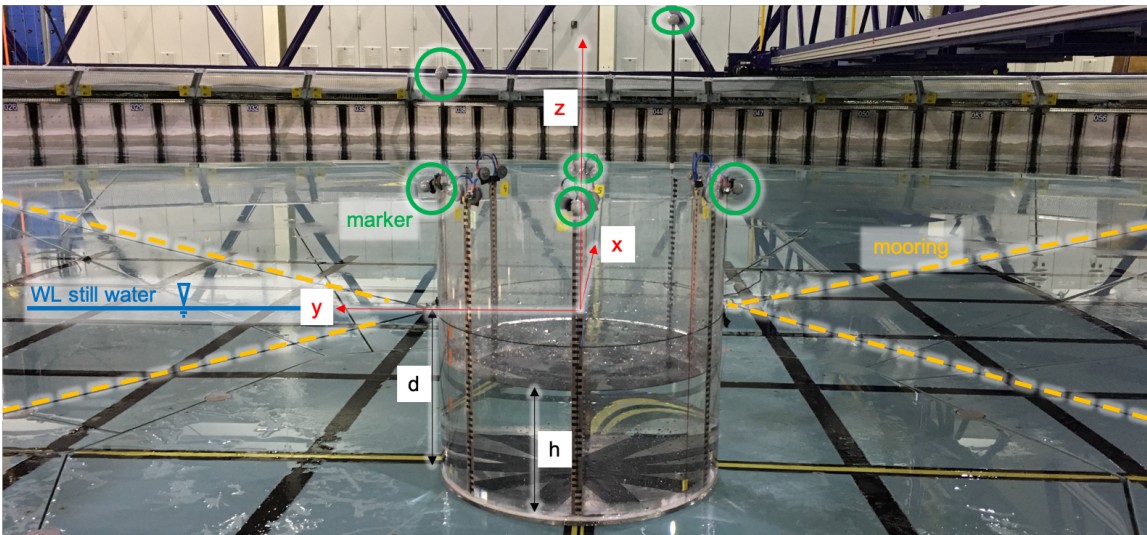

**Figure 2.** Details of the experimental set-up (variation filled with an inner water level *h*): model on the raised tank floor, highlighted mooring lines, markers for the motion capturing system and the definition of the local coordinate system.

### 2.3. Wave Gauges

As presented in Figure 1, seven wave gauges (WG) were installed on the tank's instrumentation gantry. They were aligned with the floating cylinder along the x-axis of the tank, which was the primary wave direction. Table 2 documents the x-coordinate of each gauge, and Table 3 provides the basic concept how the distances were chosen. The spacing between the WG was initially defined by a reference value *ref* of 1.27 m (available length in the tank for WG installation divided by 17, which would allow the installation of 16 WG with the same distance). The expected maximum movement of the the floating cylinder in the main wave direction, which is relatively large due to the soft moorings [13], was limiting for the exact location of the WG close to the cylinder. WG 1 was relatively close to the generating wave-makers, therefore the typical working range is in a radius of 7.5 m from the centre of the tank with a diameter of 25 m. With this chosen set-up of wave gauges, the incoming wave including the reflection at the floating cylinder was documented in a cross section. The wave gauges were calibrated at the beginning of each measurement day covering changes of the water surface of ±10 cm. The accuracy of this measurement system is typically in the order of 1 mm [17,22,23].

**Table 2.** Location of the wave gauges (WG) in relation to the global coordinate system (origin is in the centre of the tank); all *y* values are equal to 0 m.

| WGNr | WG1 | WG2 | WG3 | WG4 | WG5 | Cylinder | WG6 | WG7 |
|------|------|------|------|------|------|----------|------|------|
| x (m) | −8.215 | −5.692 | −3.168 | −1.904 | −0.630 | *var* | 3.175 | 4.443 |

**Table 3.** Distance between the wave gauges (WG) based on Table 2; spacing is based on a relation to a reference value, *ref*, of 1.27 m.

| | WG1-2 | WG2-3 | WG3-4 | WG4-5 | WG5-Center | Center-WG6 | WG5-6 | WG6-7 |
|------|-------|-------|-------|-------|------------|------------|-------|-------|
| diff (m) | 2.52 | 2.52 | 1.26 | 1.27 | 0.63 | 3.18 | 3.81 | 1.27 |
| diff/*ref* (–) | 2.0 | 2.0 | 1.0 | 1.0 | 0.5 | 2.5 | 3.0 | 1.0 |

In the following section, different analyses of the measured wave gauge data are presented. The analysis starts with the fitting of a sine function to the measurement. Those results are compared

to the intended waves sent by the wave-makers. To identify a representative wave height, different combinations of the wave gauges in front of the floating cylinder are investigated. This allows to identify for each of the eight cases a representative wave amplitude.Those values are part of the overview file described in Section 3.

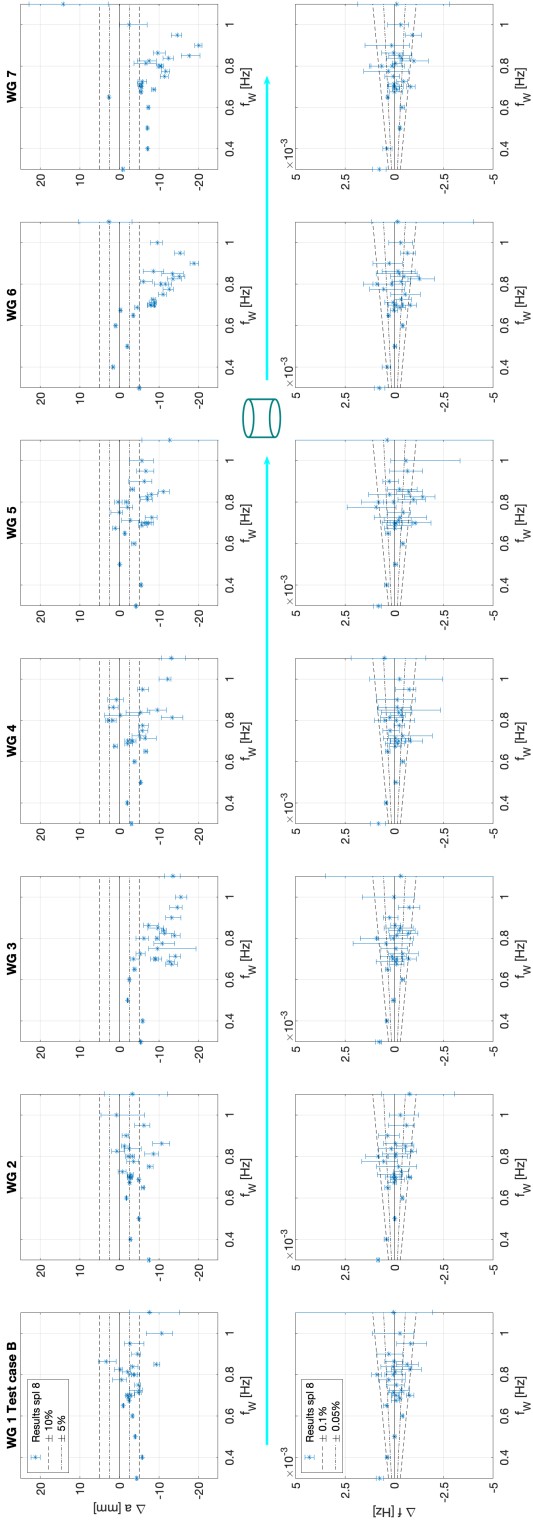

**Figure 3.** Comparison of the differences in the amplitude $\Delta a$ and frequency $\Delta f$ (measured—intended values) for all seven wave gauges (WG) in relation to the sent wave frequency $f_W$—wave direction from left to right—exemplary for test case B.

The following analysis only covers the fully developed oscillation and consequently the first 52 s after the wave-makers start are not included, similarly to the motion analysis [13]. Only the first 5 s are averaged and used to correct the zero value of each individual WG. The remaining 128 s are split in a different number of equal sub-datasets. A sine function is fitted using a least-squares cost approach, implemented in the MATLAB software package. The quality of each individual result is determined from the order of the local extreme values of the measured values as well as a quality parameter (root-mean-square error (RMSE)). Splitting into 8 constant sections delivered a good result across the investigated frequency spectrum. Splitting into 4, 6, 12, 16 and 24 segments showed no significant changes in the results or higher rate of failure for the fitting either at lower or higher frequencies. The successful fitting for the sub-datasets is averaged to deliver one result for each wave gauge and investigated wave frequency.

In a second step, the fitted sine data is compared to the intended waves. The input values for the wave-makers cover a range of the frequencies $f_W$ between 0.3 and 1.1 Hz (typical range for FloWave) and a constant amplitude $a_W$ of 50 mm. Figure 3 presents, as an example, the differences for the test case B (solid ballast option and maximal draft) for each of the seven WG. Very similar results can be found for all other seven cases. The error bar includes the range of the minimum and maximum of the sine fitting and therefore shows the biggest discrepancy before the averaging of the sub-datasets. In the upper row of the Figure 3, the difference of the amplitude $\Delta a$ is shown. Negative values indicate that the measured values are smaller than the intended wave amplitudes, which occurs for nearly the full frequency spectrum. The biggest differences can be found in the wave gauges WG6 and WG7, which are located behind the floating object near the absorbing wave-maker side. The similar analysis for the frequency shows very small differences $\Delta f$ between the sent and measured wave frequency. This indicates that no further analysis for the frequency is needed, and the use of the wave frequency $f_W$ for the presentation of the data is adequate.

Figure 4 presents different mean values of multiple wave gauges to find a representative amplitude. An average of the signal from the wave gauge in front of the cylinder is applied in the blue line (WG1-5). This value includes the very close WG to the wave-maker as well as the closest to the cylinder. Further combinations are investigated. The differences are relatively small and therefore the mean value of the wave gauges WG2-4 are chosen, which excludes the closest WG to the cylinder and the wave-maker. The minimum and maximum for this combination are also shown as dashed lines in Figure 4. In the higher frequency spectrum, this leads to obvious differences, which are caused by the fact that the receiving wave paddles struggle to absorb the high frequency waves. For the presented investigation, the peak response for heave and pitch is in a smaller frequency spectrum [13].

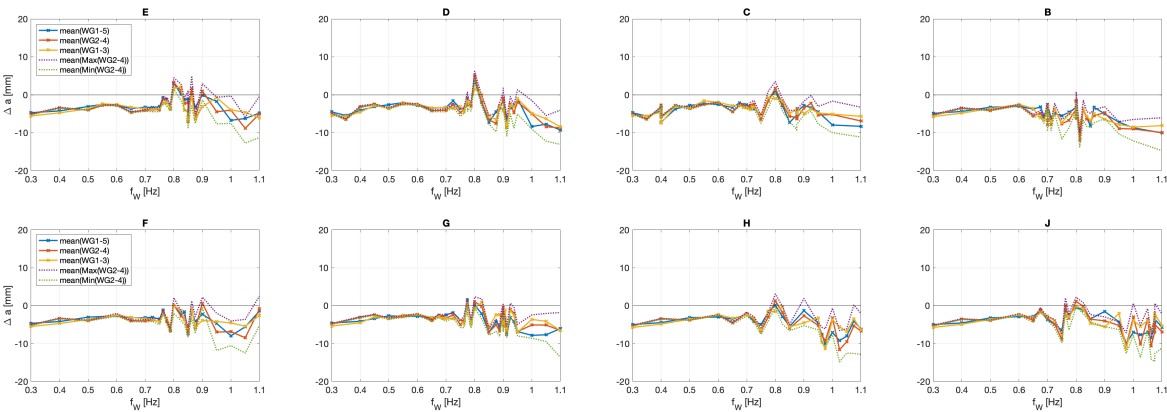

**Figure 4.** Average including different wave gauges combinations to find a representative value of the amplitude in the incoming wave field; $\Delta a$ is the difference compared to the initial amplitude of 50 mm (B, C, D, E, F, G, H, J indicate the different model configurations listed in Table 1).

In the last step, all eight investigated test cases (solid and water; four different drafts) of the measured amplitude, *a*, are combined together in Figure 5 in relation to the wave frequency, $f_W$. Up to 0.6 Hz, the differences between the variations are small. Parallel to the peak of the motion response of the floating cylinder, a slight increase can be observed in connection with a higher spread of the amplitudes. This indicates that the reflections of the tank as well as the floating cylinder are superimposed on the incoming waves. A reduction of the wave amplitude *a* from an initial value of 50 mm to 45 mm appears to be a good approach to represent the incoming wave for a validation simulation. This initial assumption can later be replaced by the full time series of the WG.

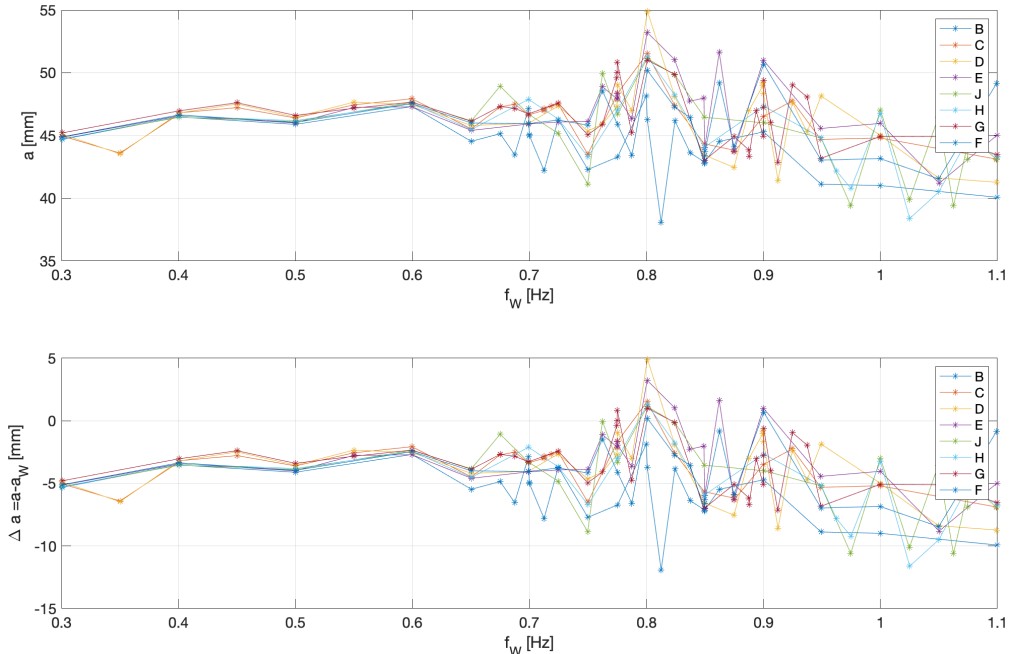

**Figure 5.** Comparison of the measured wave amplitudes *a* in relation to the wave frequency $f_W$ for all eight investigated test cases.

## 3. Dataset Description

The published dataset is available as a single ZIP container provided by the authors via the DataShare Service of the University of Edinburgh [14]. It includes two different type of files: overview and the single result. For each of the eight different cases (four drafts for solid and water ballast), one overview file is provided. The name of the file is a combination based on the following rule.

Sum_ *[case ID]* .txt

The case identification (ID) is given in Table 1, and the letters A and I are not used. Each plain text document includes two lines of header. The first line indicates the case ID and the second gives the variable description presented in Table 4. This is followed by 21–35 lines with the information of the individual runs. The first 16 runs cover the full frequency spectrum, and each model is refined with a separate number of additional experiments that was chosen on the preliminary analysis of the test results parallel to the measurement.

The first row includes the run ID, which links this input to the individual measurement file. For those in this row, the wave frequency is varied as an input to the wave-makers. This value is used to identify the individual runs in Gabl et al. [13]. The nominal amplitude for the waves for all published datasets is 50 mm, and the direction of the wave is 90°, which is parallel to the gantry (Section 1, Figure 1). The last two columns include the results of the analysis of the wave gauges presented in Section 2.3. This analysis shows that the measured frequency is very close to the targeted

value. For the normalisation of the heave movement in Gabl et al. [13], the analysed wave amplitude is used.

**Table 4.** Content of the overview files.

| Run ID | Frequency Input | Amplitude Input | Degree | Frequency Wave Gauge | Amplitude Wave Gauge |
|--------|-----------------|-----------------|--------|----------------------|----------------------|
| [−] | [Hz] | [mm] | [°] | [Hz] | [mm] |

For each individual test, the response of the floating body and wave gauges (WG) in the wave tank are available in one individual file. The name of this file includes the case and run identification (ID), starting with 01:

*[case ID] [run ID]* .txt

This plain text file includes the full time series of the measurements based on the motion capturing system (first seven columns, Section 2.2) and the seven WG (Section 2.3). Both systems record at a frequency of 128 Hz, and the synchronisation is based on the trigger of the wave-makers. In the first line of each file, the input values are repeated. The second line includes the variable name presented in Table 5. This table also shows the used units in those files. The file includes 23,040 rows of data (180 s with a frequency of 128 Hz).

**Table 5.** Content of the files for each individual run.

| X | Y | Z | Yaw | Pitch | Roll | Residual | WG1 | WG2 | WG3 | WG4 | WG5 | WG6 | WG7 |
|---|---|---|-----|-------|------|----------|-----|-----|-----|-----|-----|-----|-----|
| [mm] | [mm] | [mm] | [°] | [°] | [°] | [mm] | [mm] | [mm] | [mm] | [mm] | [mm] | [mm] | [mm] |

The provided data covers the full time of 180 s for each experimental set-up and wave frequency of the active wave-makers. Consequently, the decaying oscillation of the floating cylinder in the tank is not included. For a direct comparison of the results with a numerical simulation in the time domain, the provided time series can be reduced to the ramp-up time including the first stable oscillations [24]. This time period can be seen as nearly free floating based on the comparable soft mooring lines, and the influence of the reflections is small. For the final position of the stable oscillating model in the tank, the soft mooring not ideal as it can vary in the *x*-direction. This can potentially lead to a different influence of the reflections in the tank [13]. Future investigations will use an increased preload to limit the movement in the tank.

**Author Contributions:** R.G., T.D., E.N., J.S. and D.M.I. are responsible for the conceptualisation of the experimental investigation. R.G., T.D. and E.N. measured the data and analysed the data. R.G. and T.D. wrote the initial draft. E.N., J.S. and D.M.I. reviewed and edited the paper.

**Funding:** This work was supported by the Austrian Science Fund (FWF) under Grant J3918.

**Acknowledgments:** Open Access Funding by the Austrian Science Fund (FWF).

**Conflicts of Interest:** The authors declare no conflict of interest.

## Notation

| | |
|---|---|
| *a* | = amplitude waves (mm) |
| $\Delta a$ | = difference amplitude waves (mm) |
| *d* | = draft cylinder (m) |
| *D* | = diameter cylinder (m) |
| *f* | = frequency (Hz) |
| $f_R$ | = frequency response (Hz) |
| $f_W$ | = frequency wave (Hz) |
| $\Delta f$ | = difference frequency wave (Hz) |
| *g* | = gravity acceleration (ms$^{-2}$) |
| *h* | = water depth inside the cylinder (m) |

| | | |
|---|---|---|
| *H* | = | height of the cylinder (m) |
| WG | = | wave gauge |
| WL | = | water level |
| *x* | = | distance in the main wave direction (m) |
| *y* | = | distance orthogonal to the main wave direction (m) |
| *z* | = | distance vertical direction (m) |
| $\rho$ | = | mass density of water $\approx 998\,(\mathrm{kg\ m^{-3}})$ |

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
