# Peer review of "Experimental Data of a Floating Cylinder in a Wave Tank: Comparison Solid and Water Ballast"

_data_

Round 1
Reviewer 1 Report
The authors provided convincing explanations and made the necessary corrections and additions. In my opinion, the article can be published in its current form.
Reviewer 2 Report
one minor spelling mistake on page 2: asses should be assess.
This manuscript is a resubmission of an earlier submission. The following is a list of the peer review reports and author responses from that submission.
We want to thank the three reviewers and the editor for their work and the suggestions. The
paper was rejected with the encourage to resubmit it. Based on the positive response of at least
two reviewers and our personal opinion, we decided to resubmit the paper including the
suggested adaptations of the reviewers. This following document provides a point by point
response on the reviews. We hope that we can convince the editor and reviewers of the paper
and we are looking forward to the response. Thank you.
Round 1
Reviewer 1 Report
I recommend this paper can be merged with the contribution titled "Comparison of a Floating Cylinder with Solid and Water Ballast", Water, by the same authors to make the contribution more complete.
Authors Response:
We thank the reviewer 1 for his/her comment but we obviously disagree with the conclusion.
The water-paper focuses on the analysis of the motions and the comparison between the liquid
and solid ballast. This is one of the first steps to analyse the data. Nevertheless, this is not
essential for the validation case but generally valuable for modelling comparable structures.
Furthermore, the water paper is through the first review round with minor corrections and we
hope that the reviewer understands that we don’t want to trigger a completely new review hence
the proposed addition is a big change.
Nevertheless, in our opinion the split of the published information into two different papers is
absolutely appropriate and brings significant advantages. The main aim of the presented
research work is to provide a good data set, which will be used as a validation experiment for
floating objects under big motions. The (re-)submitted data-paper presents all the needed
information including the proposed method to quantify the exact wave amplitude. Other
researchers can pick up those measurements and analyse it differently or directly compare their
specific numerical method with the experimental observations.
We hope that the reviewer understands and support the resubmission of the paper. Thank you!
Reviewer 2 Report
Report #1 on:
Experimental Data of a Floating Cylinder in a Wave Tank – Comparison Solid and Water Ballast
by
Roman Gabl, Thomas Davey, Edd Nixon, Jeffrey Steynor, David M. Ingram
The article is logically and formally written correctly. No errors were found in this respect. The measuring set-up, the course of the experiment for various parameters and tests results were presented. A certain difficulty in getting acquainted with the research is the lack of details in the text, e.g. mooring, only reference to publication is given. Measurement data has been made available online. Basically, the article is ready for publication in its current form.
Authors Response:
We thank the reviewer for his/her assessment and agree that the mooring is only very shortly
mentioned. This is due to the fact that we included this as well as further literature reviews in
the water paper. It’s a compromise of providing key information and limit duplicates. We hope
that when both papers are available the complete needed information is available to fully use
the data as a validation experiment.
However, according to the reviewer, it would be beneficial to extend the article to:
- a detailed description and analysis of the possibilities of applying the results in real cases
Authors Response:
We understand the intention of the reviewer and see her/his point. The experiment is intended
to be a simple as possible test case to validate specific numerical codes under big motions. In
reality such cases are more likely to be found in ships and the available data in those cases is
connected with an obvious complex geometry. We tried to remove this restriction with this
experiment and allow to focus on the hydraulic effects. The water paper presents the analysis
and the comparison of the solid and water ballast. Comparable to many previous studies of
ships, the differences are comparable small but we could find an absolute interesting case,
where the cylinder starts to roll instead of pitch. Hence, we strongly believe that this is an
excellent case to test numerical codes for Fluid-Structure-Fluid-interaction. Based on such a
validation a wide range of different possible applications could be enabled to be investigated
numerically and reduce/optimise the need for further experimental work.
- the possibilities of scaling them to other issues
Authors Response:
An upscale can be conducted based on scaling laws but is not intended at the moment.
Especially the big motion response is for most applications (platforms or storing facilities) an
unwanted behaviour. But if the numerical model can cover those accurately, further adaptations
of the geometries can be conducted and the response optimised. Nevertheless, this can be an
interesting behaviour for wave energy devices. We added a short comment as well as two
further references to potential upscale of the results. We hope to cover this comment.
- a description of the range, how universal the results are, and how they relate only to a specific case
Authors Response:
Also, a very good point, thank you. It’s hard to give an exact answer to this. We varied wave
frequency with a constant requested height as well as the inner water depth and connected with
this the draft of the cylinder. In an additional measurement campaign, which is not included in
this paper, we further investigated wave angles and the exact position of the cylinder in the
tank. Those results are currently analysed but don’t show significant changes. A big uncertainty
is the diameter of the cylinder. We have not finally decided, how and if to change this but we
agree that this would be a good step forward. At the moment, we would remain on the safe side
and recommend the results as a 1:1 validation experiment. If the software can reproduce those
results including the change from pitch to roll, this would be a massive build-up of confidence
in the code/model.
- the quantity and distribution of wave gauges requires a broader description and justification
Authors Response:
We added an additional table in the manuscript, which presents the differences between the
WG and the reference value is 1.27. This is chosen by the tank diameter of 25 m, subtracted by
two times 1.75 m. This ring around the tank is not easily accessible for WG installations (safety
barrier). Those 21.5 m are divided by 17 hence we could in total install 16 WG. We did this
for a previous test and we have to admit that we left those WG installed, which fitted to the setup.
The used set-up should represent a good coverage for the incoming waves as well as behind
the cylinder.
Reviewer 3 Report
A well presented piece of work that I believe will be of interest to the wave energy community.
Authors Response:
Thank you for this response and we fully agree.
Very few comments which are minor:
Could you explain the choice of both wave amplitude and frequency? Are they based on scaled values?
Authors Response:
The frequency is chosen by the capability of the tank to deliver good wave conditions. We
previously conducted a parameter study with WAMIT to ensure that we can cover the
resonance frequency in the experimental set-up. The amplitude was fixed in the initial test run.
Obviously, we wanted to reach big motions but not sink the model. We increased the amplitude
up to a level for which we were still confident to run the model. We agree that this is not an
ideal approach but we are confident that it leaded us to the best results.
Strictly speaking the methodology should be put in the past tense.
Authors Response:
We checked and corrected this. Those changes are not specifically marked and if needed, we
can provide an additional detailed comparison. We hope that we found all cases.
Could you make figure 3 large, say a full page. There are some good results in these figures which I believe would benefit from being enlarged.
Authors Response:
We thank the reviewer for this comment and rotated the figure to use a full page for it.
_____________________________________________________________________________________
We want to thank the three reviewers and the editorial team for their valuable work and hope
that the corrected version can be supported by the reviewers as well as the editors. Thank you!